# Sutureless versus Hand-Sewn Coronary Anastomoses: A Systematic Review and Meta-Analysis

**DOI:** 10.3390/jcm11030749

**Published:** 2022-01-29

**Authors:** Marieke Hoogewerf, Jeroen Schuurkamp, Johannes C. Kelder, Stephan Jacobs, Pieter A. Doevendans

**Affiliations:** 1Department of Cardiology, University Medical Centre Utrecht, 3584 CX Utrecht, The Netherlands; j.schuurkamp1@students.uu.nl (J.S.); p.doevendans@umcutrecht.nl (P.A.D.); 2Department of Cardiothoracic Surgery, St. Antonius Hospital, 3435 CM Nieuwegein, The Netherlands; 3Department of Cardiology, St. Antonius Hospital, 3435 CM Nieuwegein, The Netherlands; keld01@antoniusziekenhuis.nl; 4Department of Cardiothoracic and Vascular Surgery, German Heart Centre Berlin, 13353 Berlin, Germany; jacobs@dhzb.de; 5Netherlands Heart Institute, 3511 EP Utrecht, The Netherlands; 6Central Military Hospital, 3584 EZ Utrecht, The Netherlands

**Keywords:** coronary artery bypass grafting, anastomosis, sutureless coronary anastomotic device, minimally invasive

## Abstract

Background: Sutureless coronary anastomotic devices are intended to facilitate minimally invasive coronary artery bypass grafting (MICS-CABG) by easing and eventually standardizing the anastomotic technique. Within this systematic review and meta-analysis, we aim to determine patency and to evaluate safety outcomes for the sutureless anastomoses. Methods: CENTRAL, MEDLINE, and EMBASE were searched from database start till August 2021 in a predefined search strategy combining the key concepts: ‘coronary artery bypass grafting’, ‘sutureless coronary anastomoses’, and ‘hand-sewn coronary anastomoses’ by the Boolean operation ‘AND’. Study characteristics, patient demographics, interventional details, and all available outcome data were extracted. A meta-analysis was performed on patency at longest follow-up. Safety outcomes were presented. Results: A total of eleven trials towards six sutureless anastomotic devices were included, comprising 3724 patients (490 sutureless and 3234 hand-sewn). There was no significant difference in patency at a mean follow-up duration of 546.3 (range 1.5–2691) days, with a risk ratio of 0.77 (95% CI 0.55–1.06). MACE was reported in 4.5% sutureless and 3.9% hand-sewn patients, including all-cause mortality (resp. 1.3 vs. 1.9%), myocardial infarction (resp. 1.6 vs. 1.7%), and coronary revascularization (resp. 1.8 vs. 0.5%). Incomplete hemostasis occurred in 24.8% of the sutureless anastomoses. Intra-operative device failure forced conversion to hand-sewn or redo-anastomosis in 5.8% of the sutureless cases. Conclusion: Based on the systematic review and meta-analysis including six devices, we conclude that sutureless coronary anastomotic devices appear safe and effective when used by well-trained and dedicated surgical teams.

## 1. Introduction

Although coronary artery bypass grafting (CABG) is one of the most frequently performed procedures within the field of cardiothoracic surgery, technical innovation seems to have virtually no place. Full sternotomy and cardiopulmonary bypass still predominate CABG cases, even though serious side effects are extensively reported. These effects include, but are not limited to, sternal dehiscence or infection, cerebrovascular events, systemic inflammatory reaction with eventual end-organ failure, and coagulation disorders [1,2].

Minimally invasive cardiac surgery (MICS-)CABG approaches intend to diminish perioperative complications and yet to provide complete and preferably total arterial revascularization to meet CABG expectations. Unfortunately, successful MICS-CABG remains reserved for the dedicated surgeons. Maneuvering robotic instruments in totally endoscopic (TE)CABG involves a long learning curve [3,4,5]. Cumbersome exposure during minimally invasive direct (MID)CABG, and hemodynamic instabilities during off-pump (OP)CABG compromise coronary anastomotic suturing and eventual completeness of revascularization [4,6,7].

To overcome the technical challenges in MICS-CABG, sutureless coronary anastomotic devices were developed. These devices intend to simplify and eventually standardize the coronary anastomotic procedure. Several sutureless devices were evaluated in the past decade, however, not one is currently routinely applied in clinical practice [4,8,9].

Since biotechnical understanding of the implantation and healing of sutureless coronary anastomotic procedures is still very limited, we aim to gain insight into the successes and failures of clinically applied devices. This information could help to objectify indications and eventual contra-indications for device usage. The objectives of this systematic review are (1) to determine patency outcomes for the sutureless coronary anastomosis, and (2) to evaluate the safety of these new sutureless procedures.

## 2. Materials and Methods

A systematic literature search was performed in accordance with the *Preferred Reporting Items for Systematic Reviews and Meta-Analyses* (PRISMA) guidelines [10]. Data extraction and analyses were performed in accordance with the *Cochrane Handbook* [11].

### 2.1. Search Strategy

CENTRAL (Cochrane), ClinicalTrials.gov, PubMed (incl. MEDLINE), and EMBASE (via OVID) were searched from database start till August 2021 for publications comparing sutureless to hand-sewn coronary anastomoses. Various search terms for ‘coronary artery bypass grafting’ were combined with terms for ‘sutureless coronary anastomoses’ and ‘hand-sewn coronary anastomoses’ by the Boolean operator “AND”. Calibration searches were performed to optimize the search strategy. The complete search is presented in the supplementary materials.

### 2.2. Eligibility Criteria

Two independent researchers (M.H. and J.S.) screened all citations yielded by the search according to a predefined protocol. Eventual disagreements were discussed and resolved by consensus with a third researcher (DS). All comparative studies, comparing sutureless to hand-sewn coronary anastomoses in CABG, were included. In vitro or animal studies and studies that included >50% emergency and/or reoperation CABG were excluded. If studies reported the same set of patients, only the most recent study was included. In order to evaluate risk of bias, we collected information on methodology for every included study (study type, method of patient selection and enrolment, way of outcome retrieval and registration, inclusion and exclusion criteria, and statistical methods). A PICOS table and the Newcastle–Ottawa quality assessment are presented in the Appendix A.

### 2.3. Data Extraction

The following baseline data were extracted from the included papers: study characteristics (date of patient enrolment, number of centers including, length of follow-up, and the number of patients included/excluded/crossed-over/lost to follow-up), patient demographics (gender, age, weight, body mass index (BMI), body surface area (BSA), hypertension, history of smoking, diabetes, peripheral artery disease, prior cerebrovascular accident (CVA) or transient ischemic attack (TIA), prior myocardial infarction (MI), chronic obstructive pulmonary disease (COPD), atrial fibrillation (AF), left ventricular function (LVF), NYHA classification, EURO score, SYNTAX score and/or number of diseased coronary arteries, prior coronary interventions, and prior cardiac surgery), and intervention details (pump strategy including eventual pump run and cross-clamp times, surgical approach including eventual conversion, target coronary artery, type of donor graft including harvesting technique, graft positioning, type of sutureless anastomotic device, hand-sewn technique used, anastomotic construction time, number of anastomoses, hemostasis, perioperative complications, and postoperative anticoagulation strategy).

All available outcome data were extracted.

### 2.4. Data Analyses

The primary outcome was patency rate at longest follow-up. Patency was determined per coronary angiography (CAG), coronary CT, or cardiac MRI.

The secondary outcomes were major adverse cardiac events (MACE) till longest follow-up and perioperative device-related complications. MACE is a composite endpoint of all-cause mortality, myocardial infarction (MI), and coronary revascularization.

Data for both primary and secondary outcomes were extracted and reported, continuous data in median (interquartile range) or mean (standard deviation), and categorical data in frequency (%). Relevant outcome data were entered into Review Manager Version 5.4 for meta-analysis. We used random effect models with generic inverse variance, presenting the risk ratio with a 95% confidence interval. Forest plots were presented, and funnel plots were examined for eventual publication bias. Heterogeneity was reported as low (I2 = 0–40%), moderate (I2 = 30–60%), substantial (I2 = 50–90%), and considerable (I2 = 75–100%). To assess the origin of heterogeneity, stratification on the length of follow-up (<3 months, or ≥3 months) was performed. Furthermore, sensitivity analyses were performed regarding the study methods of the included papers.

## 3. Results

Complete search results are presented in Figure 1. Out of 7490 titles, full text was reviewed for fourteen trials. Three trials were subsequently excluded since endpoints did not match this systematic review and/or no ethical committee approval was stated.

Eleven trials, including 3724 patients (490 sutureless vs. 3234 hand-sewn) and 4087 anastomoses (501 sutureless vs. 3586 hand-sewn) were included in the analyses [12,13,14,15,16,17,18,19,20,21,22]. Six devices for sutureless coronary anastomoses were studied: St. Jude Distal Anastomotic Device (St. Jude Inc., St. Paul, MN, USA), Magnetic Vascular Positioner (Ventrica Inc., Fremont, CA, USA), Automated Anastomotic Distal Device (Bypass Inc., Herzlia, Israel), Coronary Anastomosis Coupler (Converge Inc., Sunnyvale, CA, USA), C-port^®^ (Aesculap Inc., Central Valley, PA, USA), and U-clip^®^ (Medtronic Inc., Minneapolis, MN, USA). Device specifics are presented in Figure 2.

Study characteristics are presented in Table 1. Most, 8 out of 11, are non-randomized controlled trials (NRCT) of which 7 compare the intervention (i.e., sutureless coronary anastomosis) to an in-patient control (i.e., hand-sewn coronary anastomosis). Seven trials (287 patients) had a mean follow-up <1 year, four (3437 patients) trials had a mean follow-up >1 year. The percentage exclusions were <10% in two trials, >10 to <30% in four trials, and >30% in two trials. Three trials did not describe the number of patients excluded. Details of the exclusions are presented in Table 2. Complete extracted numbers on study design, patient demographics, interventional details, and the absence of publication bias regarding the primary endpoint are presented in the Appendix A.

### 3.1. Primary Outcome

The weighted mean follow-up duration of patency was 546.3 (range 1.5–2691) days. There was no significant difference in patency outcome between the sutureless and hand-sewn coronary anastomoses (RR 0.77; 95% CI 0.55–1.06; *p* = 0.11), as is presented in Figure 3. Subgroup analyses of each sutureless device group did not show significant differences (*p* = 0.29). However, trends can be recognized in the relatively small subgroups. The C-port studies tend to favor sutureless anastomoses (RR 0.74; 95% CI 0.51–1.06; *p* = 0.10), whereas the St Jude DAD studies tend to favor hand-sewn anastomoses (RR 15.0; 95% CI 0.91–248.21; *p* = 0.06). The overall and the C-port analyses are mainly predominated by the large sample of Balkhy et al. (66.9% of total weight) [21].

The stratification in follow-up length revealed no significant difference between sutureless and hand-sewn anastomoses in patency rates for both follow-up groups, <3 months (RR 0.56; 95% CI 0.19–1.66; *p* = 0.30) and ≥3 months follow-up (RR 0.89; 95% CI 0.55–1.44; *p* = 0.64). The forest plot is presented in the Appendix A.

Sensitivity analyses on study design also revealed no significant difference in patency between sutureless and hand-sewn anastomoses if analyzed for RCT’s only (RR 2.99; 95% CI 0.25–35.80; *p* = 0.39), while a trend in favor of sutureless anastomoses was shown for NRCT’s only (RR 0.71; 95% CI 0.51–1.00; *p* = 0.05). However, the RCT sample is a small sample, subjecting to 7.9% of the weight in total analyses. The forest plot is presented in the Appendix A.

### 3.2. Secondary Outcomes

Table 3 shows the secondary outcomes on safety and feasibility. MACE occurred in 4.5% of the patients with sutureless and 3.9% of the patients with hand-sewn coronary anastomoses. The composite endpoint includes all-cause mortality (resp. 1.3 vs. 1.9%), MI (resp. 1.6 vs. 1.7%), and coronary revascularization (resp. 1.8 vs. 0.5%). The weighted mean follow-up duration for the MACE endpoint was 808.8 (range 1.5–2691) days for the sutureless and 1090.4 (5–2691) days for the hand-sewn group.

Noteworthy are the relatively high revascularization rates in the St Jude DAD (4.8%) and the MVP group (7%). The RCT of Verberkmoes et al. presented the highest proportions of MACE (11.1 for C-port and 11.4% for hand-sewn anastomoses) in a follow-up period of 365 days [20]. Remarkably, the trials of Vicol et al. (NRCT to the MVP connector with 570 days follow-up) and Boening et al. (NRCT to the CAC connector with 60 days follow-up) both reported 0% MACE [15,18].

Incomplete hemostasis occurred in 24.8% of the sutureless anastomoses, most often reported in the C-port studies (35.3%). Intraoperative device failure, for which a redo-anastomosis or conversion to hand-sewn techniques was necessary, occurred in 5.8% of the sutureless cases. This was most often reported for the AADD device (14.3%).

## 4. Discussion

This systematic review was performed to gain insight into the clinical outcomes of sutureless coronary anastomotic devices that aim to enable MICS-CABG by simplifying the coronary anastomotic procedure. The objectives of this review were to determine patency and to evaluate safety outcomes of the clinically applied sutureless coronary anastomotic devices. We evaluated 4087 anastomoses (501 sutureless vs. 3586 hand-sewn), with the use of six different sutureless anastomotic devices.

The outcomes are:No difference in patency was determined between sutureless and hand-sewn anastomoses during a mean follow-up of almost two years.The safety evaluation reported slightly more MACEs in the sutureless than in the hand-sewn patients, mainly due to a discrepancy in coronary revascularization rates (resp. 1.8 vs. 0.5%).Feasibility of most sutureless anastomotic devices seems to be limited to specific conditions of the coronary artery target site.

Whereas overall patency outcomes did not differ between sutureless and hand-sewn anastomoses, trends can be detected between the individual sutureless coronary anastomotic devices. The St. Jude DAD showed 69.6% patency at the end of three months follow-up, whereas the U-clip resulted in 97.1% patent anastomoses over a 7-year follow-up. The latter, as an exception to the rest, for left internal mammary artery (LIMA) to left anterior descending artery (LAD) grafting. Overall patency of the hand-sewn anastomoses was 80.1%, which is in line with other trials reporting on hand-sewn saphenous vein graft (SVG) to coronary anastomoses [23,24]. Patency rates of (sutureless) coronary anastomoses are by definition determined by a complex interaction of many variables including the amount of tissue trauma induced by graft and/or coronary manipulation, the amount of non-intima surface inside the anastomosis, neo-endothelialization, and the biocompatibility of the implanted foreign material [9,25]. In addition to these anastomotic design factors, the target vessels size, the amount of graft flow, eventual flow acceleration and turbulence, and the type of donor graft used affect the ultimate patency [26,27]. The promising patency results of the U-clip could be explained by the low amount of foreign material inside the anastomosis, the intima-to-intima apposition of the donor graft and coronary artery, the use of an arterial graft, and the minimal manipulation necessary during anastomotic construction.

Safety data showed slightly more MACEs in the sutureless compared to the hand-sewn coronary anastomoses, mainly determined by higher revascularization rates (resp. 1.8 vs. 0.5%). Revascularization rates are reported to be the highest for the MVP (7%) and the St. Jude DAD (4.8%), which is in line with the presented results on patency. Myocardial infarction rates are yet quite comparable between the sutureless and hand-sewn groups (resp. 1.6 vs. 1.7%), indicating non-acute revascularization. With respect to the number of MACEs reported, one could argue that the sutureless coronary anastomoses are safe.

Safety seems to be closely related to perioperative feasibility. Considering the vast amount of excluded or converted sutureless coronary anastomoses, the coronary anatomy seems to be of uttermost importance for successful sutureless application. Most devices indicate a minimum coronary target size. Thereby, the amount of disease in mainly posterior and anterior coronary walls limits device placement, by either narrowing of the target lumen or impaired manipulation of the anterior wall. Actual device failure was reported up to 5.8% of the overall anastomoses and was highest for the AADD (14.3%) and MVP (9.4%). Loading the donor graft onto the eight circumferential pins of the AADD appeared to be complex and caused laceration of the donor graft’s tunica intima. The MVP appeared to be demanding to adequately position since the created arteriotomy should fit the magnetic planes with limited variation. In addition, incomplete hemostasis was reported in 24.8% of the sutureless anastomoses. Which was mainly an issue in the C-port (35.3%), CAC (14.6%), and St. Jude DAD (14.3%). The U-clip study was the only one without reports of perioperative device failure and hemostasis.

This systematic review is limited by the design of the included trials. Most of these trials were designed using an in-patient control. Therefore, we did not have the possibility to directly compare the safety outcome and decided to directly report this outcome rather than to perform a meta-analysis. Another limitation is the small numbers of patients included in the trials. As was reported previously, the trial of Balkhy et al. predominates the sample. Meta-regression analysis was also not considered possible due to the small samples included.

The results of our review on patency and safety are in line with the previous, presented pooled series of sutureless coronary anastomotic devices [8,9]. No new devices are clinically introduced since, yet new comparative studies with the two market approved devices were published and included in this review [21,22]. While C-port production was discontinued since 2020, U-clip now is the only market approved sutureless anastomotic device. Notwithstanding the safety and efficacy of the devices, device costs and complicated device handling can prevent a device from market entrance. Especially the exclusions of the evaluated studies, presented in Table 2, show the delicate balance in constructing standardized anastomoses on a varying coronary anatomy. Most devices require prespecified ranges in vessel wall thickness and diameter, emphasizing the current need for proper patient selection. Thereby, only two of the included studies reported on minimally invasive access techniques. Four out of six studies administered dual antiplatelet therapy (DAPT) instead of single Aspirin. The combination of these conditions and device costs that outrange the regular suture, discourage the use of sutureless coronary anastomotic devices.

The current evaluation, however, represents first-generation devices. We recommend clinical evaluation with known approaches as sternotomy and six months of DAPT postoperatively for those devices that add extra foreign material to the anastomotic surface. Yet, future generations should adapt towards minimal access strategies and reduce device limitations in a rapid pace. Preclinical research is currently performed for three new devices: the S2 Connector (iiTech, Amsterdam, The Netherlands), the ELANA^®^ Heart Bypass (AMT Medical, Ede, The Netherlands), and a new device for coronary suture fixation (researcher driven) [28,29,30]. In particular, the ELANA Heart Bypass shows a completely new approach, by first anastomosing both vessels and only thereafter creating the arteriotomy per contact laser. This ensures a proper fit between anastomosis and arteriotomy, standardizes the arteriotomy, and avoids intima-trauma.

The aim to improve coronary revascularization by enabling MICS-CABG procedures and to lower the threshold for total arterial revascularization is fundamental for the development of sutureless coronary anastomotic devices. Based on this concept, the devices are intended to simplify and eventually standardize the coronary anastomotic procedure. This review emphasizes that these sutureless coronary anastomotic procedures appear to be safe and effective when used by well-trained and dedicated surgical teams.

## 5. Conclusions

Based on the published data of six devices, we conclude that sutureless coronary anastomotic devices appear to be safe and effective. Yet, when used by dedicated surgical teams under predefined anatomical conditions, these devices could potentially enable MICS-CABG procedures and lower the threshold for total arterial revascularization. Notwithstanding the encouraging results, only one device is currently on the market. However, three novel devices are in preclinical evaluation.

## Figures and Tables

**Figure 1 jcm-11-00749-f001:**
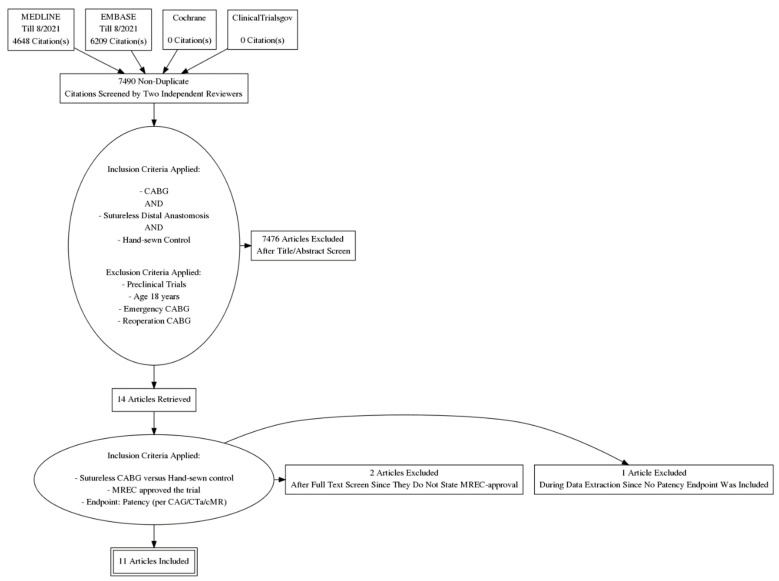
PRISMA flow-chart of systematic review search outcomes. CABG: coronary artery bypass grafting; CAG, coronary angiography; MREC: medical research ethics committee; CTa: computed tomography angiography; cMR: cardiac magnetic resonance imaging.

**Figure 2 jcm-11-00749-f002:**
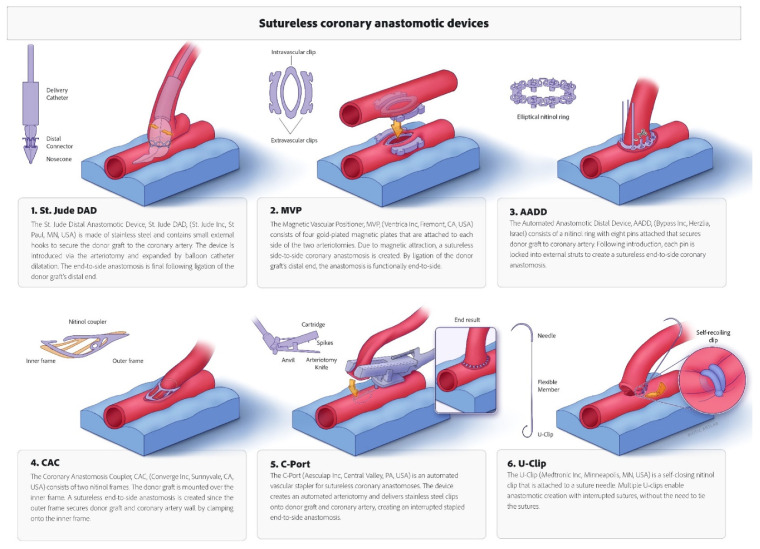
Overview of the sutureless coronary anastomotic devices included in the systematic review.

**Figure 3 jcm-11-00749-f003:**
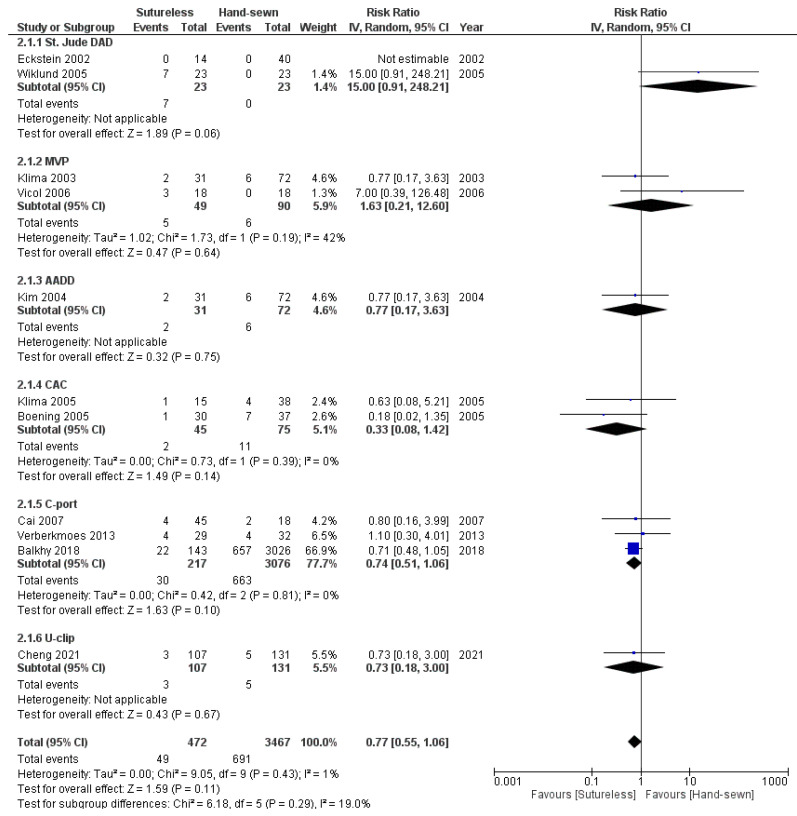
Forest plot on anastomotic patency, presented per device subgroup. Line of no effect represented by vertical line. Outcome measures for individual studies are presented per box (risk ratio) and horizontal line (95% confidence interval). The summary outcome per device group and for the total group is presented in a diamond shape (risk ratio and 95% confidence interval).

**Table 1 jcm-11-00749-t001:** Study characteristics.

Study	Device	Method	*N*	Age	Male	OPCAB	Approach	Graft	Target	Anticoagulation	*N* Anastomoses		FU-Time
			S/HS	S/HS	S/HS	S/HS					S	HS	In Days
Eckstein 2002 [12]	St. Jude DAD	NRCT	14	63 (8)	-	0 (0%)	-	SVG	non-LAD	-	14	40	60
Wiklund 2005 [13]	St. Jude DAD	RCT	30/30	69 (48–83)/67 (42–81) ^a^	95%/85%	0 (0%)	-	SVG	non-LAD	Aspirin for life	28	30	179
Klima 2003 [14]	MVP	NRCT	32	65 (9)	27 (90%)	6 (19%)	-	IMA, SVG	All	DAPT once	32	74	6.3 (5.4)
Vicol 2006 [15]	MVP	NRCT	11	-	10 (91%)	-	-	IMA, SVG, RA	All	-	18	18	570 (105)
Kim 2004 [16]	AADD	NRCT	14	65 (7)	10 (71%)	14 (100%)	sternotomy	IMA, RGEA, SVG	All	-	14	34	1.5 (1.3)
Klima 2005 [17]	CAC	NRCT	15	65.9 (8.6)	13 (87%)	0 (0%)	-	SVG	non-LAD	-	15	38	657 (146)
Boening 2005 [18]	CAC	NRCT	46	63.3 (7.5)	44 (96%)	0 (0%)	sternotomy	SVG	non-LAD	DAPT for 28 days	33	81	60
Cai 2007 [19]	C-Port	NRCT	50	68.0 (9.7)	-	46 (92%)	-	SVG	non-LAD	DAPT for 3 months	69	46	90
Verberkmoes 2013 [20]	C-Port	RCT	35/36	67.6 (5.6)/66.5 (5.7)	31 (89%)/32 (89%)	5 (17%)/3 (9%)	all approaches	SVG	non-LAD	Aspirin for life	35	36	345 (27)
Balkhy 2018 [21]	C-Port	HC	117/3014	-	-	116 (99%)/-	-	IMA, SVG	All	-	117	3026	390
Cheng 2021 [22]	U-clip	NRCT	126/154	59.1 (9.0)/60.7 (10.2)	105 (83%)/120 (77.9%)	126 (100%)/154 (100%)	TECAB/RADCAB	IMA	LAD/All	DAPT for life	126	163	2691 (912)

Abbreviations: S, sutureless coronary anastomosis group; HS, hand-sewn coronary anastomosis group; OPCAB, off-pump coronary artery bypass grafting; NRCT, non-randomized controlled trial; RCT, randomized controlled trial; HC, historical control group; TECAB, totally endoscopic coronary artery bypass; RADCAB, robotically assisted direct coronary artery bypass; SVG, saphenous vein graft; IMA, internal mammary artery; RGEA, right gastroepiploic artery; RA, radial artery; LAD, left anterior descending artery; DAPT, dual anti-platelet therapy; FU, follow-up. Data reported in mean (SD) or N (%) or otherwise as described; ^a^ described in mean (range). - data not reported.

**Table 2 jcm-11-00749-t002:** Exclusions.

	Method	*N* Total S/HS	*N* Excluded S/HS	Reason Exclusion
Eckstein 2002 [12]	NRCT	14	9	9× outer diameter coronary target <3.0 mm
Wiklund 2005 [13]	RCT	30/30	2/0	1× conversion for plaque in the posterior wall 1× conversion for insufficient graft length
Klima 2003 [14]	NRCT	32	9	1× conversion for plaque in the posterior wall 3× conversion for inadequate hemostasis 5× not inserted for improper judgement of coronary target size
Vicol 2006 [15]	NRCT	11	-	NA
Kim 2004 [16]	NRCT	14	5	2× failure of graft positioning onto connector pins 2× small and diseased coronary target 1× conversion to ECC
Klima 2005 [17]	NRCT	15	18	18× no consent for follow-up CAG
Boening 2005 [18]	NRCT	46	13	8× exclusion for graft diameter < 3 mm, coronary target diameter < 2 mm, or heavy calcification of the coronary target 1× conversion for too fragile coronary target 3× conversion for bleeding 1× conversion for low flow
Cai 2007 [19]	NRCT	50	-	NA
Verberkmoes 2013 [20]	RCT	35/36	4/3	7× did not meet the intraoperative inclusion criteria
Balkhy 2018 [21]	HC	117/3014	27/-	23× did not meet the intraoperative inclusion criteria 4× conversion
Cheng 2021 [22]	NRCT	126 /154	-/-	NA

Abbreviations: *N*, number of patients; S, sutureless coronary anastomosis group; HS, hand-sewn coronary anastomosis group; NRCT, not-randomized controlled trial; RCT, randomized controlled trial; HC, historical control; ECC, extra-corporal circulation; CAG, coronary angiography; NA, not applicable. - data not reported. Conversion is a conversion from sutureless to hand-sewn coronary anastomosis unless stated otherwise.

**Table 3 jcm-11-00749-t003:** Safety outcomes.

	During Follow-Up					During Hospital Admission		
Study	Follow-Up, In Days	MACE	Mortality	MI	Revascularization	SAE	Incomplete Haemostasis	Device Failure
Eckstein 2002 [12]	60	1/14 (7.1%)	0/14 (0%)	0/14 (0%)	1/14 (7.1%)	-	0/14 (0%)	0/14 (0%)
Wiklund 2005 [13]	180	1/28 (3.6%) vs. 1/30 (3.3%)	0/28 (0%) vs. 1/30 (3.3%)	0/28 (0%) vs. 0/30 (0%)	1/28 (3.6%) vs. 0/30 (0%)	1 reoperation for bleeding (S) 1 CVA (HS)	6/28 (21.4%)	0/28 (0%)
Subtotal St Jude DAD		2/42 (4.8%)	0/42 (0%)	0/42 (0%)	2/42 (4.8%)		6/42 (14.3%)	0/42 (0%)
Klima 2003 [14]	30	4/32 (12.5%)	1/32 (3.1%)	1/32 (3.1%)	3/32 (9.4%)	3 reoperations for bleeding (S) 1 prolonged ventilation (S) 1 TIA (S)	0/32 (0%)	3/32 (9.4%)
Vicol 2006 [15]	570	0/11 (0%)	0/11 (0%)	0/11 (0%)	0/11 (0%)	-	-	-
Subtotal MVP		4/43 (9.3%)	1/43 (2.3%)	1/43 (2.3%)	3/43 (7%)		0/32 (0%)	3/32 (9.4%)
Kim 2004 [16]	1.5	-	0/14 (0%)	-	-	-	1/14 (7.1%)	2/14 (14.3%)
Subtotal AADD			0/14 (0%)				1/14 (7.1%)	2/14 (14.3%)
Klima 2005 [17]	-	-	-	-	-	-	3/15 (20%)	-
Boening 2005 [18]	60	0/33 (0%)	0/33 (0%)	0/33 (0%)	0/33 (0%)	-	4/33 (12.1%)	0/33 (0%)
Subtotal CAC		0/33 (0%)	0/33 (0%)	0/33 (0%)	0/33 (0%)		7/48 (14.6%)	0/33 (0%)
Cai 2007 [19]	90	1/50 (2%) vs. 4/193 (2.1%)	1/50 (2%) vs. 3/193 (1.6%)	0/50 (0%) vs. 1/193 (0.5%)	0/50 (0%) vs. -	2 reoperations for bleeding (S) 6 reoperations for bleeding (HS)	-	8/50 (16%)
Verberkmoes 2013 [20]	365	4/35 (11.4%) vs. 4/36 (11.1%)	1/35 (2.9%) vs. 1/36 (2.8%)	2/35 (5.7%) vs. 2/36 (5.6%)	1/35 (2.9%) vs. 1/36 (2.8%)	1 prolonged ventilation (S) 2 prolonged inotropics or IABP (S) 1 reoperation for bleeding (S) 1 prolonged ventilation (HS) 2 prolonged inotropics or IABP (HS) 1 reoperation for bleeding (HS) 1 reoperation for ischemia (HS)	22/35 (62.9%)	4/35 (11.4%)
Balkhy 2018 [21]	pre discharge	5/117 (4.3%) vs. -	1/117 (0.9%) vs. -	4/117 (3.4%) vs. -	0/117 (0%) vs. -	-	45/155 (29%)	4/155 (2.6%)
Subtotal C-port		10/202 (5%)	3/202 (1.5%)	6/202 (3%)	1/202 (0.5%)		67/190 (35.3%)	16/240 (6.7%)
Cheng 2021 [22]	2691	4/126 (3.2%) vs. 7/154 (4.5%)	2/126 (1.6%) vs. 3/154 (1.9%)	0/126 (0%) vs. 4/154 (2.6%)	2/126 (1.6%) vs. 0/154 (0%)	1 reoperation for bleeding (S) 8 wound infections (HS)	-	-
Subtotal U-clip		4/126 (3.2%)	2/126 (1.6%)	0/126 (0%)	2/126 (1.6%)			
Total hand-sewn		16/413 (3.9%)	8/413 (1.9%)	7/413 (1.7%)	1/222 (0.5%)			
Total sutureless		20/446 (4.5%)	6/460 (1.3%)	7/446 (1.6%)	8/446 (1.8%)		81/326 (24.8%)	21/361 (5.8%)

All events are presented: event count/total count (%) for the sutureless group vs. hand-sewn group respectively. -, no data available. Incomplete hemostasis and device failure are presented for the device-anastomoses only, excluded anastomoses excluded. Abbreviations: MACE, Major Adverse Cardiac Events, composite endpoint of all-cause mortality, myocardial infarction, and revascularization; MI, myocardial infarction; SAE, serious adverse events; CVA, cerebrovascular accident; TIA, transient ischemic attack; IABP, intra-aortic balloon pump; (S) event occurred in sutureless-group; (HS) event occurred in hand-sewn group.

## Data Availability

Data is contained within this article and Appendix A.

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
