# Peer review of "Sutureless versus Hand-Sewn Coronary Anastomoses: A Systematic Review and Meta-Analysis"

_jcm, 2022, doi:10.3390/jcm11030749_

Round 1
Reviewer 1 Report
Authors should be congratulated for their work. Few comments to improve the quality of the manuscript.
1) A PICOS table for inclusion / exclusion criteria could be helpful.
2) Quality assessment of the papers (e.g. Newcastle Ottawa scale,...) should be included in supplementary materials.
3) Secondary endpoints should be analyzed with Forest plot.
4) Publication bias should be assessed for all endpoints (it was only included in methods but not in results).
5) Considering the number of trial and the extensive work made by authors, a meta-regression analysis could be helpful. Alternatively, please discuss its limitations in your paper.
Author Response
Reviewer 1
Authors should be congratulated for their work. Few comments to improve the quality of the manuscript.
Dear reviewer, Thank you for your time and the quality added to our work.
1)
Comment: A PICOS table for inclusion / exclusion criteria could be helpful.
Answer: Thank you for your suggestion. We added a PICOS table to the supplementary materials.
2)
Comment: Quality assessment of the papers (e.g. Newcastle Ottawa scale,...) should be included in supplementary materials.
Answer: Following up on your excellent comment, we added the quality assessment according to the Newcastle Ottowa scale to the supplementary materials.
3)
Comment: Secondary endpoints should be analysed with Forest plot.
Answer: Due to the study design of most included studies, presenting an in-patient control, it is not possible to analyse the secondary endpoint with a Forest plot. This, since the secondary endpoint includes patients, not individual anastomoses. We added paragraph five to the discussion to explain this limitation.
4)
Comment: Publication bias should be assessed for all endpoints (it was only included in methods but not in results).
Answer: Publication bias was assessed for the outcome patency, since this outcome was included in meta-analysis. We now added the funnel plot to the supplementary data.
5)
Comment: Considering the number of trial and the extensive work made by authors, a meta-regression analysis could be helpful. Alternatively, please discuss its limitations in your paper.
Answer: Thank you. We do, however, not have the possibility for meta-regression analysis considering the small sample sizes included. We did now add this information to the extra paragraph five in the discussion.

Reviewer 2 Report
The authors present a well written review on sutureless conoronary anastomosis devices, which have mostly been used in studies in the past with only one device being currently available on the market.
The review is comprehensive and certainly of interest to the reader given the lack off novel techniques in the field of CABG surgery.
The authors should add further information on anticoagulation requirements in the postoperative period for these devices.
Given the diameter requirements for coronary arteries, the authors should further discuss for what percentage of patients undergoing CABG are these devices an option and ist this cost-effective as compared to hand-sutured anastomosis? In small and/or severely calcified vessels, are these devices useful?
Further, the authors argue that these devices allow or potentially simplify minimially invasive (MICS) CABG procedures. The authors should add data on how many patients in the cited manuscripts underwent MICS procedures and further discuss surgical approach options and feasibility of device utilization for the different coronary arteries.
Author Response
Reviewer 2
The authors present a well written review on sutureless coronary anastomosis devices, which have mostly been used in studies in the past with only one device being currently available on the market.
The review is comprehensive and certainly of interest to the reader given the lack of novel techniques in the field of CABG surgery.
Dear reviewer, Thank you for your time and insightful comments.
1)
Comment: The authors should add further information on anticoagulation requirements in the postoperative period for these devices.
Answer: Thank you for your insightful comment. We now briefly discussed the anticoagulation requirements for these devices in the discussion, paragraph 7-8. The anticoagulation strategies used by the included studies are presented in table 1.
2)
Comment: Given the diameter requirements for coronary arteries, the authors should further discuss for what percentage of patients undergoing CABG are these devices an option and is this cost-effective as compared to hand-sutured anastomosis? In small and/or severely calcified vessels, are these devices useful?
Answer: Thank you for this excellent comment. We now evaluated this issue more extensively in the discussion, paragraph 7-8.
3)
Comment: Further, the authors argue that these devices allow or potentially simplify minimally invasive (MICS) CABG procedures. The authors should add data on how many patients in the cited manuscripts underwent MICS procedures and further discuss surgical approach options and feasibility of device utilization for the different coronary arteries.
Answer: Following up on your comment, we did remark this subject in the discussion, paragraph 7-8. The numbers itself are presented in table 1.

Round 2
Reviewer 1 Report
authors should be congratulated for the revised version of their manuscript.